# Predictors of Frequent Emergency Department Use and Hospitalization among Patients with Substance-Related Disorders Recruited in Addiction Treatment Centers

**DOI:** 10.3390/ijerph19116607

**Published:** 2022-05-28

**Authors:** Marie-Josée Fleury, Zhirong Cao, Guy Grenier, Christophe Huỳnh

**Affiliations:** 1Douglas Hospital Research Centre, Douglas Mental Health University Institute, 6875 LaSalle Blvd, Montreal, QC H4H 1R3, Canada; marie-josee.fleury@douglas.mcgill.ca (M.-J.F.); zhirong.cao@douglas.mcgill.ca (Z.C.); guy.grenier@douglas.mcgill.ca (G.G.); 2Department of Psychiatry, McGill University, 1033, Pine Avenue West, Montreal, QC H3A 1A1, Canada; 3Centre Intégré Universitaire de Santé et Des Services Sociaux du Centre-Sud-de-l’Île-de-Montréal, Institut Universitaire sur les Dépendances, 950 Louvain Est, Montreal, QC H2M 2E8, Canada; christophe.huynh.ccsmtl@ssss.gouv.qc.ca

**Keywords:** substance-related disorders, addiction treatment centers, frequent emergency department use, hospitalization, patient sociodemographic and clinical characteristics, patterns of service use

## Abstract

Few studies have assessed the overall impact of outpatient service use on acute care use, comparing patients with different types of substance-related disorders (SRD) and multimorbidity. This study aimed to identify sociodemographic and clinical characteristics and outpatient service use that predicted both frequent ED use (3+ visits/year) and hospitalization among patients with SRD. Data emanated from 14 Quebec (Canada) addiction treatment centers. Quebec administrative health databases were analyzed for a cohort of 17,819 patients over a 7-year period. Multivariable logistic regression models were produced. Patients with polysubstance-related disorders, co-occurring SRD-mental disorders, severe chronic physical illnesses, and suicidal behaviors were at highest risk of both frequent ED use and hospitalization. Having a history of homelessness, residing in rural areas, and using more outpatient services also increased the risk of acute care use, whereas high continuity of physician care protected against acute care use. Serious health problems were the main predictor for increased risk of both frequent ED use and hospitalization among patients with SRD, whereas high continuity of care was a protective factor. Improved quality of care, motivational, outreach and crisis interventions, and more integrated and collaborative care are suggested for reducing acute care use.

## 1. Introduction

Patients with substance-related disorders (SRD), including substance-induced or substance use disorders, substance intoxication or withdrawal [1], are often reported as frequent emergency department (ED) users (3+ ED visits/year) and many are hospitalized [2,3,4]. Frequent ED use and hospitalization are recognized as key indicators of adverse outcomes [5]. These services are also very costly, and their use often indicates a lack of access to other health services [6]. According to a US study, patients with substance use disorders accounted for 9.4% of all ED use and 11.9% of hospitalization between 2014 and 2018 [7]. A 2009–2010 Swiss study [8] found that 35% of frequent ED users had substance use disorders, while a recent Canadian study identified hospitalization rates of 24% among patients with SRD who used ED [9]. Many patients with SRD are affected by co-occurring SRD-mental disorders (MD) [10] or by co-occurring SRD-chronic physical illnesses (e.g., cardiovascular diseases), which also increases the odds of frequent ED use and hospitalization [11]. 

While most patients with SRD using acute care (ED, hospitalization) may require outpatient care, particularly SRD treatment, few receive this care [12]. ED and inpatient services are thus key settings where patients may be screened, provided with brief motivational intervention [7], and referred to more appropriate resources, facilitating access to care [13]. Identifying patient characteristics and patterns of outpatient service use associated with frequent ED use and hospitalization may thus inform the development of appropriate alternatives for the care of patients with SRD.

Few studies have assessed predictors of acute care use among patients with SRD who access specialized SRD treatments offered by addiction treatment centers [4,12]. In Quebec, these services treat about 10% of the population with SRD [14], representing some of the most vulnerable patients. However, no study, to our knowledge, has assessed the impact of overall outpatient service use on frequent ED use and hospitalization, comparing patients with different types of SRD and multimorbidity. This study aimed to identify patient sociodemographic and clinical characteristics and outpatient service use that predicted frequent ED use and hospitalization for any medical reason among a large cohort of patients with SRD recruited in Quebec addiction treatment centers. We hypothesized that patients with more serious health problems would have a higher risk of acute care use, and that using SRD programs in addiction treatment centers and having high continuity of physician care would protect against acute care use. 

## 2. Materials and Methods

### 2.1. Study Context–the Quebec Healthcare System 

The Canadian province of Quebec has a public healthcare system. Specialized services for SRD are provided by regional addiction treatment centers. These centers offer treatment programs such as detoxification, substitution or reintegration treatments and brief intervention units. SRD services are accessible through self-referral, referral from primary care services or by court order. Co-occurring SRD-MD are usually treated in specialized psychiatric services, mostly provided in hospital settings. These specialized services are complementary to primary care services, including the care provided by general practitioners (GP) working mainly in family medicine groups, or by psychosocial teams (e.g., social workers, psychologists) working in community healthcare centers. Family medicine groups are GP clinics where patients must register, and where nurses and social workers enhance services by ensuring continuity of patient care and extended medical coverage. 

### 2.2. Study Sample, Design, and Sources

Data for this study emanated from 14 of the 16 Quebec addiction treatment centers. Study data were gathered for a cohort of 18,228 patients with SRD registered in the addiction treatment center database (SIC-SRD) during the 2012–2013 financial year. SIC-SRD data were available from 1 April 2009 to 31 March 2016. Patients entered into the cohort had to be Quebec residents, 12+ years old, and eligible for the Quebec Health Insurance Plan (Régie de l’assurance maladie du Québec, RAMQ) between 2009–2010 and 2015–2016 (1 April–31 March). Patients were excluded if they died during the study period or were hospitalized 91+ days in 2014–2015, the year before measurements for acute care use (the study outcomes) were taken, which would have precluded appropriate measurement of their outpatient service use. 

The two outcome variables, frequent ED use and hospitalization, were measured in the 2015–2016 financial year. Independent variables were grouped as follows: sociodemographic characteristics measured in 2015–2016, in the most recent year, or from 2009–2010 to 2015–16 (i.e., criminal history or history of homelessness); clinical characteristics from 2012–2013 to 2014–2015; and outpatient service use characteristics within 12 months prior to the third ED visit or first hospitalization in 2015–2016, except for dropouts from addiction treatment centers, whose measures were taken between 2009–2010 and 2014–2015. Each patient without frequent ED use and hospitalization in 2015–2016 was allocated the same exposure window as a randomly selected patient with the same age and sex, and from the same type of residential area, who made an ED visit. Figure 1, the conceptual framework for the study, identifies all databases linked to each study variable and their measurement intervals. Data from all the databases were merged every year for each patient through a unique RAMQ identifier matched with the SIC-SRD database.

Data from addiction treatment centers (SIC-SRD) included patient sociodemographic characteristics, SRD diagnoses, and services received in these centers. The RAMQ integrated billing systems for most physician services, excluding 6% of services that occurred outside the public system [15]. RAMQ included various sub-databases (e.g., hospitalization, ED use, psychosocial interventions in community healthcare centers). Diagnostic codes from RAMQ were framed by the International Classification of Diseases Ninth and Tenth Revisions (Appendix A). As the study used administrative data from provincial administrative health databases (RAMQ, SIC-SRD), patient informed consent was not required. Access to the databases was granted by the Quebec Commission for Access to Information, and an ethics review board of a health and social services organization approved the study protocol.

### 2.3. Variables

The two outcome variables, frequent ED use and hospitalization, were measured for any medical reason. Frequent ED use was defined as 3+ visits/year, the standard designation for frequent ED use [16,17]. This usually reflects the inadequacy of services in response to patient needs or indicates that quality of care is inappropriate for responding to patient health conditions [5]. Considering independent variables, sociodemographic characteristics included sex, age group, living situation (e.g., alone), principal occupation (e.g., unemployed), material and social deprivation, type of residential area (e.g., urban), criminal history, and history of homelessness. The Material Deprivation Index, based on the smallest dissemination areas established for the 2011 Canadian census, integrated ratios of population employment, average income and education levels less than high school, while the Social Deprivation Index included proportions of patients living alone, those with single civil status (single, separated, divorced or widowed) and single-parent families [18]. Both indices were classified in quintiles but were regrouped for this study into three categories representing less (1–2), moderate (3) and more (4–5) deprived areas, or areas not assigned for patients without an address, e.g., homeless individuals and nursing home residents.

Clinical characteristics included type of SRD, type of MD, suicidal behaviors (suicide ideation and attempt), chronic physical illnesses, and SRD only or with co-occurring disorders (i.e., exclusive group: co-occurring SRD-MD only, co-occurring SRD-chronic physical illnesses only, and co-occurring SRD-MD-chronic physical illnesses). SRD referred to alcohol and drug-related disorders (induced or use disorders, intoxication, withdrawal). 

Type of SRD, referring to exclusive groups, included alcohol only, cannabis only, drugs other than cannabis (e.g., cocaine, opioids) only, and polysubstance-related disorders. Type of MD referred to schizophrenia spectrum and other psychotic disorders, bipolar disorders, personality disorders, anxiety or depressive disorders, adjustment disorders, attention deficit/hyperactivity disorder, and other MD. If a patient had more than one MD, the most serious MD (following the above order) was identified as the principal MD. Chronic physical illnesses, based on an adapted version of the Elixhauser Comorbidity Index [19], included level of severity per patient from 0–3+. Each chronic physical illness has a specific weight for severity (e.g., ulcer disease = 0; chronic pulmonary disease = 1). The rate (0–3+) indicated the total level of severity for all chronic physical illnesses affecting patients. 

Outpatient service use characteristics included usual physician (usual GP or usual psychiatrist only, both usual GP and psychiatrist, and no usual physician), frequency of consultations with usual GP or psychiatrist, high continuity of physician care, frequency of psychosocial interventions received in community healthcare centers (excluding GP consultations), or in any SRD treatment program in addiction treatment centers, and percentage of patient dropouts from SRD programs in addiction treatment centers. Usual GP, a proxy for family doctor, was defined as having at least two consultations with the same GP working in private clinics or community healthcare centers, or with at least two GP working in the same family medicine group. Usual psychiatrist was defined as one who followed a patient in outpatient care at least twice. Alternatively, patients who made only one consultation with a psychiatrist had to have consulted their GP at least twice, which was a proxy for collaborative care [20]. Highest frequency of care was defined as at least 4+ consultations or interventions/year [21,22,23]. Continuity of physician care was measured with the Usual Provider Continuity Index [24], which is one of the most common indices used to measure continuity of care, on a score from 0 to 1 [25]. The Index describes the proportion of consultations with the usual GP or psychiatrist of all GP and psychiatrists consulted in outpatient care, including consultations in walk-in clinics. A score of ≥0.80 is considered high continuity of care [24]. The percentage of patient dropouts from SRD programs represented the number of SRD programs that patients had discontinued on their own for all the episodes of treatment received by patients in these centers prior to finalization of their treatment plans. 

### 2.4. Analyses 

Descriptive analyses were performed for all study variables. Missing values were less than 1%, and complete case analysis was used [26]. The intraclass correlation coefficient (ICC) for the study was small (<0.01), indicating low shared variance among patients from the 14 addiction treatment centers and precluding the need for multilevel analysis. Bivariate logistic regression was used to examine the associations between each independent variable and both outcomes, frequent ED use and hospitalization. Sensitivity analyses were produced by testing several independent variables, especially those related to outpatient service use, which were measured in the prior 12, 6 or 3 months of acute care, and for which providers were regrouped, or not (e.g., GP service use only, both GP and psychiatrist). All these sensitivity analyses yielded similar results. Based on criterion procedures for forward model selection, independent variables identified as significant in the bivariate analyses (alpha set at 0.20) were entered into the multivariable logistic regression models. The Akaike Information Criterion (AIC) [27] was used to select independent variables for the final multiple logistic models, and collinearity statistics were tested using variance inflation factors (VIF), with five as the maximum level of VIF [28]. Odds ratios (OR), *p*-values and 95% confidence intervals were reported. The statistical analyses were performed using the STATA SE (version 17) program [29].

## 3. Results

Of the initial sample of 18,228 patients, 331 died and 78 were hospitalized for 91+ days during the measurement period for outpatient service use and were excluded from the study. Of the 17,819 patients in the final sample, 18% made frequent use of ED, and 17% were hospitalized in 2015–2016. Regarding patient characteristics, 66% were men, 45% were 25–44 years old, 56% lived in more materially deprived areas and 62% in more socially deprived areas, while 52% resided in urban areas, and 14% had a history of homelessness (Table 1). Results for diagnoses showed 64% of patients with polysubstance-related disorders, 73% with MD, including 24% with schizophrenia spectrum and other psychotic disorders or bipolar disorders; 16% with suicidal behaviors; 41% with chronic physical illnesses; 40% with co-occurring SRD-MD only, and 34% with SRD-MD-chronic physical illnesses. In the 12-month period prior to acute care use, 48% of patients had no usual outpatient physician and 36% a usual GP only, while 44% had high continuity of physician care; 38% received psychosocial interventions from community healthcare centers and 28% had interventions from addiction treatment centers, 40% of which had high drop-out rates over the 6-year study period.

Compared with patients living in urban areas, those from rural areas had 17% greater odds of frequent ED use and 23% greater odds of hospitalization; whereas, living in semi-urban areas decreased the odds of frequent ED use by 20% but increased the odds of hospitalization by 13% (Table 2). Individuals with a history of homelessness had 50% greater odds of frequent ED use than other patients with SRD, and 22% greater odds of hospitalization. Patients in the 12–17 and 18–24 age groups and those living in more materially deprived areas had 47, 37 and 29% greater odds of frequent ED use, respectively, compared with patients 45+ years or those living in less materially deprived areas. Women had 14% greater odds of hospitalization than men.

Compared with patients affected by cannabis-related disorders only, patients affected by drug-related disorders other than cannabis only and those with polysubstance-related disorders had 61 and 83% greater odds of frequent ED use, respectively, while patients with alcohol-related disorders only and polysubstance-related disorders had 33 and 59% greater odds of hospitalization. Patients with suicidal behaviors had 2.57 times greater odds of frequent ED use and 1.53 greater odds of hospitalization, than patients without these behaviors. Compared to patients with an index level-0 for severity of chronic physical illness, those with levels 1, 2 and 3 had 77, 98 and 4.23 times greater odds of frequent ED use, as well as 1.21, 1.88 and 10.64 times greater odds of hospitalization. Having co-occurring SRD-MD, SRD-chronic physical illnesses, and SRD-MD-chronic physical illnesses amplified the odds of frequent ED use by 93, 46 and 1.77 times, respectively, as compared with having SRD only, while the odds of hospitalization for these same diagnoses increased 93-, 82-, and 1.89-fold. 

Patients with a usual GP only, usual psychiatrist only, or both usual GP and psychiatrist had 39, 74 and 56% greater odds of being frequent ED users, while their risks of hospitalization were 39%, 1.64 times, and 76% greater than patients without a usual physician. However, patients with high continuity of physician care had 27% lower odds of frequent ED use and 20% lower odds of hospitalization. Patients receiving 1–3 or 4+ psychosocial interventions in community healthcare centers had 62 and 77% greater odds of frequent ED use, and 31 and 52% greater odds of hospitalization than patients without psychosocial interventions. Compared to patients with low dropout rates from addiction treatment centers, those with median and high dropout rates had 30 and 25% greater odds of frequent ED use, respectively.

## 4. Discussion

The proportion of frequent ED users (18%) in this study was similar to results reported in a 2012–2013 Canadian clinical study [30], which identified 20% of frequent ED users, with frequent ED use defined as 5+ visits/year. The proportion of hospitalized patients with SRD (17%) in this study was somewhat lower than that reported in another Canadian clinical study at 24% [9]. As expected, patients were highly vulnerable, most experiencing material or social deprivation, living alone and unemployed. The majority were also affected by polysubstance-related disorders, mainly cannabis associated with other drug- and alcohol-related disorders. Most had either co-occurring SRD-MD (40%) or SRD-MD-chronic physical illnesses (34%). Despite poor overall health, most patients in the cohort lacked a usual physician or psychiatrist, or follow-up at addiction treatment centers or community healthcare centers over the 12-month study period. 

The findings confirmed the first hypothesis that patients with more serious health problems would have higher risk of acute care use. Compared to patients with no chronic physical illness and SRD only, the risks were elevated among patients with severe chronic physical illnesses (index 3+), four times higher for frequent ED use and roughly 11 times higher for hospitalization, while among patients with co-occurring SRD-MD-chronic physical illnesses, the risk was nearly double for acute care service use. Patients with SRD presenting suicidal behaviors had nearly 2 to 3 times greater odds of hospitalization and frequent ED use, respectively, than those without these behaviors. Around one fourth of patients in this cohort had serious MD, a result considerably higher than the 5.6% reported in a recent US population survey [31]. Patients with serious MD are recognized as frequent acute care users [32,33]. Studies have shown that SRD, particularly polysubstance-related disorders (64% of the sample), increase the risk of co-occurring MD [34] and severe chronic physical illnesses [35], contributing to acute care use [36]. Suicidal behaviors were also elevated in the cohort at around 16%. A 2015 US population study found that 12-month rates for suicide ideation and attempt were 4% and 0.6%, respectively [37]. Previous studies also reported generally strong associations between SRD, MD and suicidal behaviors [38], contributing to acute care use [9,39]. A 2014–2015 Canadian study found than 50% of frequent ED users with SRD presented with suicidal behaviors [9]. Patients with SRD and severe co-occurring chronic physical illnesses, MD, and suicidal behaviors could thus benefit from better screening, provision of brief motivational intervention and referral to various resources, according to their needs. 

Compared to patients with cannabis-related disorders only, it is unsurprising that those with polysubstance-related disorders had more than 2.5 times higher odds of acute care use, as associated SRD increases the odds of poorer overall health [35]. Higher risk of frequent ED use among patients with drug-related disorders other than cannabis only may result in greater risk of intoxication or exposure to overdose from these drugs [40,41] given their popularity in recreational contexts and their appeal for improving energy or performance, but this also leads to accidents or aggressive behavior [42]. Increased implementation of harm reduction strategies such as drug checking [43], safer supply services [44] and supervised injection sites [45] may help diminish these adverse outcomes. As for higher risk of hospitalization among patients with alcohol-related disorders exclusively, compared to those with cannabis-related disorder only, this may be explained by the high prevalence of co-occurring chronic physical illnesses for these patients [46]. Alcohol-related disorders commonly lead to high mortality risk [47]. 

The second hypothesis was partially confirmed, as higher dropout rates from addiction treatment centers over the 6-year period, compared with lower rates (below 34%), predicted higher risk of frequent ED use. Only a minority of patients (28%) were still using addiction treatment centers within the 12-month study period for outpatient services (prior to acute care measurement in 2015–2016), occurring 3 years after patient recruitment in these centers. Most patients (61%) had a median or high percentage of dropout from SRD programs over the 6-year study period. Elevated service dropout is a widespread problem among patients with SRD, whose overall underutilization of outpatient services is mainly explained by the difficulties of creating therapeutic alliances [48], forced abstinence [49], issues of stigmatization [50], and dissatisfaction with services [51]. Motivational interventions [52], patient-centered approaches to care, and programs more responsive to patient needs [53] may be deployed more effectively to improve patient retention in programs and increase outpatient service use. 

The third hypothesis was also confirmed, as high continuity of physician care, a well-known indicator of quality care [54], did emerge as a protective factor against acute care use. However, fewer than half of patients (44%) received high continuity of physician care, suggesting the need for improvements given the high level of multimorbidity in this sample. High continuity of care may promote therapeutic alliances and access to integrated care and was associated with better medication adherence in a previous study [55]. Moreover, greater use of outpatient care was associated in this study, as in previous research [12,13], with more acute care use, a result that may simply reflect the service seekers’ great need for additional healthcare services, or the fact that the frequency of care was insufficient. Issues around the quality of treatment, another well-known problem among patients with SRD and MD, may also be at play [56,57], and should be addressed. In Quebec, few integrated SRD-MD treatment programs or services dispensing intensive case management or assertive community services exist for patients with SRD, nor has collaborative care between the social and healthcare sectors or between various providers been strongly deployed [58]. 

It was interesting to see that patients with SRD at higher risk of frequent ED use and hospitalization were also most vulnerable in terms of their living conditions. Homelessness is recognized as a life situation frequently associated with SRD [59] that leads to frequent use of acute care services [4,7]. Many homeless individuals also had mental disorders or chronic physical illnesses, and lacked appropriate outpatient care [60]. Poor material conditions are also associated with more recurrent hospitalization among patients, as identified in previous studies [12,59]. Compared with patients residing in urban areas, those living in rural areas were more at risk for acute care use, which likely related to the lack of outpatient services in these areas, as reported elsewhere [61]. Regarding the lower risk of frequent ED use in semi-urban areas compared with urban areas, it was possible that suburban area patients used urban services, which are often more accessible, and may allow them to disguise their health conditions. The urban services may also have been closer to the workplaces of these patients. However, the risk of hospitalization was higher in semi-urban versus urban areas. Greater risk of frequent ED use among patients in the 12–24 age group, compared with those 45+, may have resulted from their high-risk behaviors (e.g., binge drinking, car accidents) [62,63]. Underutilization of outpatient services is also more common among youth [64]. As in our study, other research found that women were more at risk of hospitalization than men [4,65], perhaps corresponding to their relatively higher prevalence rates for MD and chronic physical illnesses [66].

Some limitations in this study must be noted. First, as health administrative databases were primarily developed for financial or management purposes, not research, they provide only proxy measures of patient needs. Second, while frequent ED use was defined as 3+ visits/year, other thresholds would have generated different results. Third, some clinical conditions such as trauma or acute illnesses were not considered in the study, despite the potential of these conditions to increase acute care. Fourth, data on other services that may have helped prevent acute care use, such as Alcoholics or Narcotics Anonymous, community-based organizations such as crisis centers, or hospital psychosocial services, were not available in the Quebec administrative databases. Fifth, data relating to other quality issues or types of interventions such as harm reduction or the implementation of best practices could not be assessed with the study database. Finally, these findings may not be generalizable to healthcare systems too divergent from the Quebec system, particularly those without universal coverage.

## 5. Conclusions

This study was innovative in identifying patient characteristics and patterns of outpatient service use that predicted acute care use for a large cohort of patients with SRD recruited in addiction treatment centers. The findings revealed that patients with polysubstance-related disorders, co-occurring SRD-MD, and severe chronic physical illnesses, as well as patients presenting with suicidal behaviors, were at greatest risk for both frequent ED use and hospitalization. Having a history of homelessness, residing in rural areas and being high outpatient service users also increased the risk for acute care use, whereas high continuity of physician care protected against acute care use. Lower dropout rates from SRD programs in addiction treatment centers also protected against frequent ED use. To decrease use of acute care, flexible long-term follow-up care including crisis services, integrated services for SRD-MD-chronic physical illness and collaborative care need to be more effectively implemented. Improving motivational and outreach interventions as well as harm reduction strategies to increase patient retention, especially in addiction treatment center services, should also be emphasized. Finally, as patients attending addiction treatment centers are a highly vulnerable population, they should all have at least a usual physician to ensure a high continuity of care and coordination with critical psychosocial resources. 

## Figures and Tables

**Figure 1 ijerph-19-06607-f001:**
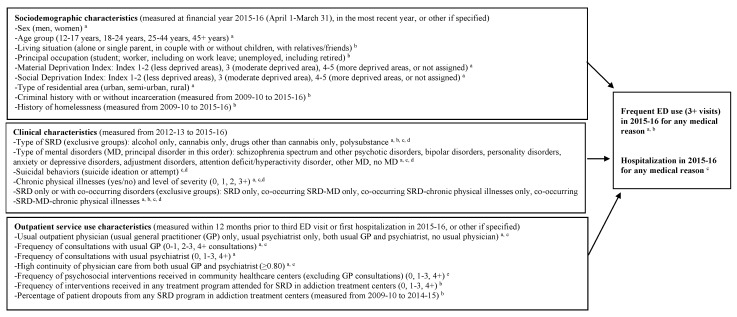
Conceptual framework: predictors of frequent emergency department (ED) use and hospitalization among patients with substance-related disorders (SRD). ^a^ *Régie de l’assurance maladie du Québec* (RAMQ, physician database); ^b^ *Système d’information clientèle pour les services de réadaptation dépendances* (SIC-SRD, addiction treatment center database, including SRD diagnostics based on standardized instruments); *^c^ Banque de données communes des urgences* (BDCU, ED database); ^d^
*Maintenance et exploitation des données pour l’étude de la clientèle hospitalière* (MED-ECHO, hospitalization database); ^e^ *Système d’information permettant la gestion de l’information clinique et administrative dans le domaine de la santé et des services sociaux* (1-CLSC, psychosocial interventions in community healthcare centers, including GP working on salary). For definitions of variables included in the study see footnotes for the tables in Section 3 or the Section 2. Details on diagnostic codes and instruments are presented in Appendix A.

**Table 1 ijerph-19-06607-t001:** Characteristics of patients using addiction treatment centers (*n* = 17,819, or other if specified).

	*N*	%
**Sociodemographic characteristics** (2015–2016, in the most recent year, or other if specified)		
Men	11,676	65.53
Women	6143	34.47
Age group (years)		
12–17	800	4.49
18–24	2960	16.61
25–44	8016	44.99
45+	6043	33.91
Living situation (*n* = 16,134)		
Alone (or single parent)	7376	45.72
Couple with or without children	3121	19.34
Living with relatives/friends	5637	34.94
Principal occupation ^a^		
Student	2422	13.59
Worker, including on work leave	5756	32.30
Unemployed, including retired	9641	54.11
Material Deprivation Index ^b^		
1 and 2	4748	26.65
3	3114	17.48
4, 5 and not assigned	9957	55.88
Social Deprivation Index ^b^		
1 and 2	4029	22.61
3	2755	15.46
4, 5, and not assigned	11,035	61.93
Type of residential area (*n* = 17,802)		
Urban (>100,000)	9241	51.91
Semi-urban (10,000 to 100,000)	5193	29.17
Rural (<10,000)	3368	18.92
Criminal history with or without incarceration (2009–2010 to 2015–2016)	3467	19.46
History of homelessness (2009–2010 to 2015–2016)	2417	13.56
**Clinical characteristics** (2012–2013 to 2015)		
Type of substance-related disorders (SRD, exclusive groups)		
Alcohol only	3451	19.37
Cannabis only	1575	8.84
Drugs other than cannabis only	1379	7.74
Polysubstance	11,414	64.06
Cannabis and other drugs ^c^	1957	10.98
Cannabis and alcohol ^c^	1325	7.44
Drugs other than cannabis and alcohol ^c^	3967	22.26
Cannabis, other drugs and alcohol ^c^	4165	23.37
Type of mental disorders (MD, principal disorder) ^d^		
Schizophrenia spectrum and other psychotic disorders	2584	14.50
Bipolar disorders	1617	9.07
Personality disorders	2084	11.70
Anxiety or depressive disorders	5465	30.67
Adjustment disorders	439	2.46
Attention deficit/hyperactivity disorder	423	2.37
Other MD	409	2.30
No MD	4798	26.93
Suicidal behaviors (suicide ideation or attempt)	2779	15.60
Chronic physical illnesses ^e^	7278	40.84
Elixhauser comorbidity index		
0	14,504	81.40
1	1056	5.93
2	1191	6.68
3+	1068	5.99
SRD only or with co-occurring disorders (exclusive groups)		
SRD only	3500	19.64
Co-occurring SRD-MD only	7041	39.51
Co-occurring SRD-chronic physical illnesses only	1298	7.28
Co-occurring SRD-MD-chronic physical illnesses	5980	33.56
**Outpatient service use characteristics** (within 12 months prior to third emergency department (ED) visit or first hospitalization in 2015–2016, or other if specified) ^f^		
Usual outpatient physician ^g^		
Usual general practitioner (GP) only	6338	35.57
Usual psychiatrist only	1226	6.88
Both usual GP and psychiatrist	1754	9.84
No usual physician	8501	47.71
Frequency of consultations with usual GP ^g^		
0–1	9727	54.59
2–3	4139	23.23
4+	3953	22.18
Frequency of consultations with usual psychiatrist ^g^		
0	14,839	83.28
1–3	1347	7.56
4+	1633	9.16
High continuity of physician care from both usual GP and psychiatrist (≥0.80) ^h^	7823	43.90
Frequency of psychosocial interventions received in community healthcare centers (excluding GP consultations) ^i^		
0	10,999	61.73
1–3	3523	19.77
4+	3297	18.50
Frequency of interventions received in any treatment programattended for SRD in addiction treatment centers ^j^		
0	12,811	71.90
1–3	1523	8.55
4+	3485	19.56
Percentage of patient dropouts from any SRD program in addiction treatment centers (2009–2010 to 2014–2015) ^k^		
Low (0 to 33%)	6867	38.54
Median (34 to 66%)	3800	21.33
High (67 to 100%)	7152	40.14
**Outcomes** (2015–2016)		
Frequent ED use (3+ visits) for any medical reason ^l^	3221	18.08
Hospitalization for any medical reason	3018	16.94

^a^ This included exclusive groups, representing the principal occupation of patients at the time when measurements were taken. For example, it is possible that a patient classified as a student was also working part-time. ^b^ Material and social deprivation indices are related to the smallest residential dissemination areas, based on the 2011 Canadian census. For this study, quintiles were regrouped into three levels representing less (1–2), moderate (3) and more (4, 5, or not assigned) deprived areas. “Not assigned” areas related to missing address or living in an area where index assignment was not feasible. An index cannot usually be assigned to residents of nursing homes or to homeless individuals (see Section 2 for more information). ^c^ Variables included in descriptive analyses only. ^d^ If a patient had more than one MD, the most severe MD was identified as her/his “principal MD”. We also considered “validity” of the diagnosis, by selecting MD diagnosed more than once, during a hospitalization, or by the patient’s usual physician, especially the psychiatrist. MD severity was considered in this order: (1) schizophrenia spectrum and other psychotic disorders, (2) bipolar disorders, (3) personality disorders, (4) anxiety or depressive disorders, (5) adjustment disorders, (6) attention deficit/hyperactivity disorder, (7) other MD. Details on the diagnostic codes are presented in Appendix A. ^e^ Chronic physical illnesses included: renal failure, cerebrovascular illnesses, neurological illnesses, endocrine illnesses, tumor without or with metastasis, chronic pulmonary illnesses, diabetes complicated and uncomplicated, cardiovascular illnesses, and other chronic illness categories (e.g., blood loss anemia) (see Appendix A for the complete list of chronic physical illnesses, definition of the index and referencing Method). ^f^ Each patient without any ED visit in 2015–2016 was allocated the same exposure window as a randomly selected patient with the same age and sex, and from the same type of residential area, who made an ED visit (see Section 2). ^g^ Usual outpatient physicians are those who ensure continuity of care. Usual general practitioner (GP) is a proxy for “patient family physician”. To be considered as having a usual GP, the patient had to have at least two consultations with the same GP, or at least two consultations with GP working in the same family medicine group, as defined in the Section 2. Usual psychiatrist was defined as one that followed any patient in outpatient care at least twice. Alternatively, patients who made only one outpatient consultation with a psychiatrist had to have consulted their GP at least twice, which was considered a proxy for collaborative care (see references in Section 2). ^h^ Continuity of physician care is measured with the Usual Provider Continuity Index, describing the proportion of consultations with the usual GP or psychiatrist of all GP and psychiatrists consulted in outpatient care (including consultations in walk-in clinics). A score ≥ 0.80 is considered high continuity of care. References are provided in Section 2. ^i^ Community healthcare centers provide mainly psychosocial interventions delivered through multidisciplinary teams (e.g., social workers, nurses, psychologists). These services are thus complementary to the care provided by GP, and both are primary care (or first line) services. ^j^ Treatment programs offered in addiction treatment centers included: medical activities (e.g., substitution treatment), specialized addiction services, either internal (e.g., detoxification treatment) or external (e.g., counseling, rehabilitation), and brief treatment (see Section 2). ^k^ The addiction treatment database (SIC-SRD) provided reasons justifying patient case closure (e.g., treatment dropout, treatment completion, patient relocation to another area not covered by the center). Percentage of patient dropouts from SRD programs represent the number of SRD programs that patients had discontinued on their own for all the episodes of treatment received by patients in these centers prior to finalization of their treatment plans. It was possible to calculate the percentage of dropouts per patient, accounting for all programs used by the patient over the 6-year data collection period. ^l^ A minimum of three visits per year is the standard definition for frequent ED use, based on previous research. References are provided in the Section 2.

**Table 2 ijerph-19-06607-t002:** Multivariable logistic regression results among patients with substance-related disorders (SRD) with frequent emergency department (ED) use, or hospitalization in 2015–2016.

	Frequent ED Use (3+)				Hospitalization			
	OR	*p*-Value	95% CI		OR	*p*-Value	95% CI	
**Sociodemographic characteristics** (2015–2016 or in the most recent year)								
Women (ref.: men)					1.14	0.005	1.04	1.26
Age group (ref.: 45+ years)								
12–17	1.47	0.001	1.17	1.86				
18–24	1.37	<0.001	1.18	1.58				
25–44	1.10	0.065	0.99	1.22				
Material Deprivation Index (ref.: 1 and 2) ^a^								
3	1.05	0.468	0.92	1.21				
4, 5, and not assigned	1.29	<0.001	1.16	1.43				
Type of residential area (ref.: urban (>100,000))								
Semi-urban (10,000 to 100,000)	0.80	<0.001	0.72	0.89	1.13	0.023	1.02	1.25
Rural (<10,000)	1.17	0.008	1.04	1.31	1.23	0.001	1.09	1.38
History of homelessness (2009–2010 to 2015–2016)	1.50	<0.001	1.34	1.68	1.22	0.001	1.08	1.38
**Clinical characteristics** (2012–2013 to 2015–2016)								
Type of SRD (exclusive groups, ref.: cannabis-related disorders only)								
Alcohol only	1.20	0.138	0.94	1.52	1.33	0.016	1.05	1.68
Drugs other than cannabis only	1.61	<0.001	1.24	2.08	1.03	0.823	0.78	1.36
Polysubstance	1.83	<0.001	1.48	2.26	1.59	<0.001	1.28	1.97
Suicidal behaviors (suicide ideation or attempt)	3.57	<0.001	3.24	3.95	2.53	<0.001	2.28	2.81
Chronic physical illnesses (Elixhauser comorbidity index (ref.: 0)) ^b^								
1	1.77	<0.001	1.50	2.10	2.21	<.001	1.87	2.61
2	1.98	<0.001	1.69	2.32	2.88	<0.001	2.47	3.36
3+	5.23	<0.001	4.45	6.15	11.64	<0.001	9.88	13.72
SRD only or with co-occurring disorders (exclusive groups, ref.: SRD only)								
Co-occurring SRD-mental disorders (MD) only	1.93	<0.001	1.63	2.28	1.93	<0.001	1.59	2.34
Co-occurring SRD-chronic physical illnesses only	1.46	0.002	1.14	1.86	1.82	<0.001	1.42	2.34
Co-occurring SRD-MD-chronic physical illnesses	2.77	<0.001	2.31	3.33	2.89	<0.001	2.36	3.54
**Outpatient service use characteristics** (within 12 months prior to third ED visit or first hospitalization in 2015–2016, or other if specified) ^c^								
Usual outpatient physician (ref.: no usual physician) ^d^								
Usual general practitioner (GP) only	1.39	<0.001	1.19	1.61	1.39	<0.001	1.18	1.63
Usual psychiatrist only	1.74	<0.001	1.43	2.12	2.64	<0.001	2.16	3.22
Both usual GP and psychiatrist	1.56	<0.001	1.31	1.87	1.76	<0.001	1.46	2.11
High continuity of physician care score from both usual GP and psychiatrist (≥0.80) ^e^	0.73	<0.001	0.64	0.84	0.80	0.002	0.69	0.92
Frequency of psychosocial interventions received in community healthcare centers (excluding GP consultations) (ref.: 0) ^f^								
1–3	1.62	<0.001	1.46	1.80	1.31	<0.001	1.17	1.47
4+	1.77	<0.001	1.59	1.97	1.52	<0.001	1.36	1.70
Percentage of patient dropouts from any SRD program in addiction treatment centers (2009–2010 to 2014–2015) (ref.: low (0 to 33%)) ^g^								
Median (34 to 66%)	1.30	<0.001	1.16	1.45	1.11	0.091	0.98	1.24
High (67 to 100%)	1.25	<0.001	1.13	1.38	1.04	0.428	0.94	1.16

^a^ Material and social deprivation indices are related to the smallest residential dissemination areas, based on the 2011 Canadian census. For this study, quintiles were regrouped into three levels representing less (1–2), moderate (3) and more (4, 5, or not assigned) deprived areas. “Not assigned” areas related to missing address or living in an area where index assignment was not feasible. An index cannot usually be assigned to residents of nursing homes or to homeless individuals (see Section 2 for more information). ^b^ Chronic physical illnesses included: renal failure, cerebrovascular illnesses, neurological illnesses, endocrine illnesses, tumor without or with metastasis, chronic pulmonary illnesses, diabetes complicated and uncomplicated, cardiovascular illnesses, and other chronic illness categories (e.g., blood loss anemia) (see Appendix A for the complete list of chronic physical illnesses, definition of the index and referencing Method). ^c^ Each patient without any ED visit in 2015–2016 was allocated the same exposure window as a randomly selected patient with the same age and sex, and from the same type of residential area, who made an ED visit (see Section 2). ^d^ Usual outpatient physicians are those who ensure continuity of care. Usual general practitioner (GP) is a proxy for “patient family physician”. To be considered as having a usual GP, the patient had to have at least two consultations with the same GP, or at least two consultations with GP working in the same family medicine group, as defined in the Section 2. Usual psychiatrist was defined as one that followed any patient in outpatient care at least twice. Alternatively, patients who made only one outpatient consultation with a psychiatrist had to have consulted their GP at least twice, which was considered a proxy for collaborative care (see references in Section 2). ^e^ Continuity of physician care is measured with the Usual Provider Continuity Index, describing the proportion of consultations with the usual GP or psychiatrist of all GP and psychiatrists consulted in outpatient care (including consultations in walk-in clinics). A score ≥ 0.80 is considered high continuity of care. References are provided in Section 2. ^f^ Community healthcare centers provide mainly psychosocial interventions delivered through multidisciplinary teams (e.g., social workers, nurses, psychologists). These services are thus complementary to the care provided by GP, and both are primary care (or first line) services. ^g^ Treatment programs offered in addiction treatment centers included: medical activities (e.g., substitution treatment), specialized addiction services, either internal (e.g., detoxification treatment) or external (e.g., counseling, rehabilitation), and brief treatment (see Section 2).

## Data Availability

In accordance with the applicable ethics regulations in the province of Quebec, the authors do not have permission to share the data extracted from the Quebec Health Insurance Board (RAMQ) database for this study.

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
