# Peer review of "Predictors of Frequent Emergency Department Use and Hospitalization among Patients with Substance-Related Disorders Recruited in Addiction Treatment Centers"

_ijerph, 2022, doi:10.3390/ijerph19116607_

Round 1
Reviewer 1 Report
Review for IJERPH
Date: 01/05/2022
Paper: Predictors of frequent emergency department use and hospitalization among patients with substance-related disorders recruited in addiction treatment centers
Overall points
This paper addresses important considerations and useful findings for continuity of care for patients with substance use disorders and for healthcare system spending. The paper highlights the protective nature of outpatient care for these patients and confirms a wealth of literature on the risks for frequent emergency department use. However, I found the paper very difficult to follow; many aspects of the methodology could be described more clearly. The paper is very similar to previous papers published by the authors (i. Variables associated with low, moderate and high emergency department use among patients with substance-related disorders and ii. Frequencies of emergency department use and hospitalization comparing patients with different types of substance or polysubstance-related disorders), with the main differences being that the first paper utilised a longer timeframe with the same datasets and an extra dependent variable (hospitalisation), while paper 2 used very similar data and time frames. It may be useful to describe how this paper builds on the previous work.
Comments on the paper by section
Abstract
I found the first sentence of the abstract confusing and it was only as I read the paper that I understood what the authors were trying to say here. I suggest rewording the sentence.
Introduction
- In lines 39 and 40, the authors state that “patients with SRD accounted for 9.4% of all ED use and 11.9% of acute care hospitalization between 2014 and 2018” Please could they clarify if this includes people using substances at risky levels, but who do not meet criteria for a substance use disorder according to DSM-5. I see that the ICD includes “intoxication”. It would be useful to state that this is covered in the term “substance-related disorders” since not all readers would be familiar with this. Please also make this distinction when mentioning other data. In lines 40/41, a meta-analysis is referenced to back up the statement that patients with SRD “used ED 4.8 times more on average, and were hospitalized 7.1 times more than the general population”, but this paper only includes studies with data on participants who use illicit drugs. The data for this paper includes patients admitted to addiction treatment centres so I’m assuming that they had more severe disorders (probably aligned with DSM-5 substance use disorders, rather than just “intoxication”) so does it make sense to discuss evidence relating to people using substances at hazardous levels? There is a great deal of ED data on patients engaging in risky drinking/other substance use but they are a different population to patients accessing services at addiction treatment centres.
- Line 45: what is “SRD-chronic physical illnesses”? Do the authors mean physical illnesses related to the substance use or just SRD comorbid with physical illnesses?
- Lines 48 and 49 don’t seem to be referring to patients with SRD and don’t seem to add much to the flow of the argument. I suggest deleting.
- Lines 54 and 55: “Identifying … patterns of outpatient service use associated with frequent ED use and hospitalization” doesn’t state the idea clearly. It would make more sense to me from a health system point of view, to talk about “identifying characteristics of acute care users with SRD, and the possible gaps in their non-acute healthcare service access in order to prevent further acute care use.” Lines 60 and 61 are also a bit awkward and unclear. I suggest rewording.
Study purpose and rationale
- The study investigated acute care use for chronic medical illnesses. Why were surgical/trauma/acute illnesses excluded?
- Line 67: “using SRD programs in addiction treatment centers … would protect against acute care use” – but I thought all study cases had accessed addiction treatment centres. (In the discussion, this hypothesis is stated more clearly. Please align.) Or are the authors comparing those who dropped out vs those who did not drop out? What was the definition of “drop out”? (Regarding timing, outcome etc. If the person had become abstinent and were doing well, would they still have counted as a “drop out”?)
Methodology
Study context
- Please clarify which services are provided where – the sentence spanning lines 75-78 is ambiguous. Please address each service location separately. This description of the healthcare system will be very helpful when clarified slightly.
- Minor point: line 80 – should be “ensure”, not “insure”.
Study sample
I became tangled in the text under 2.2. It would be helpful to separate out the descriptions of each database. Then how they were used. Figure 1 was helpful and the study design would have been clearer if I had looked at that figure first. The description of the variables also helped but this was only described further down in the paper. The appendix with the codes was helpful.
- When mentioning the time frame for the data accessed, it would be clearer to say from the 1st of April 2009 to the 31st of March 2016. I first read this as data were only accessed for the years 2009/2010 and 2015/2016. Although, I am not sure if data from the years 2009-2011 were used. Figure 1 suggests these data weren’t used.
- Patients were identified from the 2012/2013 SIC-SRD data, and data from all the other databases were accessed from 2009 to 2016? Outpatient service use data were accessed for 12 months before 2015/2016 3rd ED visit or hospitalization.
- Lines 90-92 contain an incomplete sentence – it’s not clear what was done with those databases.
- Fig 1: the lower 2 textboxes have cut off some of the letters in the last line.
Variables
- This makes sense to me – it sounds like the exposure window refers to the outpatient service use. Correct? Maybe clarify. Are you using a case-control study design? “Each patient without frequent ED use and hospitalization in 2015-16 was allocated the same exposure window as a randomly selected patient with an ED visit with the same age and sex, and from the same type of residential area.”
- Lines 110-111: I was lost again with the dropouts. Was this about the outpatients service use data?
- The Social and Material Deprivation Indices: does it make sense to include “not assigned” with the more deprived areas. I understand that “not assigned” could include homeless people/people without a fixed address. Were there other reasons for “not assigned”? I see nursing homes are also mentioned in the footnotes of Table 1. A bit more information on the Social Deprivation Index would be useful. May be a good idea to provide the information in the methods rather than in long footnotes.
- Elixhauser Comorbidity Index: please clarify what was used in the analysis – number or severity. Or are these combined in the index? The authors mention number of comorbidities, as well as severity. How is severity captured in the index?
- The description of the outpatient service use variables was quite involved but makes sense, and it seems like this was measured in the 12 months prior to the 3rd ED visit or to hospitalisation. Then in the analysis section: “Several independent variables were tested, especially those related to outpatient service use which were measured in the prior 12, 6 or 3 months of acute care outcomes, and for which providers were regrouped, or not (e.g., GP service use only, both GP and psychiatrist), which had all similar results.” Now there are more time frames in the mix – 3 and 6 months as well?
- It would be good to have more information about the Usual Provider Continuity Index, where it is used and what the limitations are of this using this index. How does the index deal with 0 or 1 consultations?
Ethical considerations
Understandably, the data is not freely available. Regarding the Quebec Health Insurance Board, are beneficiaries aware that their data is used to monitor services, used for reports/publications etc? If this is in place, it would be good to mention.
Results
- The authors mention that patients were excluded due to death or prolonged hospital stays in 2014/2015. How many were excluded due to missing data? (The authors mention complete case analysis.)
- Line 183: drop-out rates were high. Is it possible that some of these patients were abstinent or no longer met criteria for an SRD? Would you expect all patients with SRD irrespective of clinical trajectory to remain in care?
- Table 1: there are 4 types of SRD mentioned in the variables section. There are 4 other categories included in Table 1 – with the note that these were included in the descriptive analysis only.
- Table 1: Please explain this variable “Percentage of patient dropouts from any SRD program in addiction treatment centers (2009-10 to 2014-15)”. Is this the number of times (frequency) patients dropped out of a program? It would help to reword.
- Table 2: Please add ref for “women vs men” as per other variables in the table. The text describing Table 2 jumps up and down the table and between frequent ED use and hospitalization. It would be good to implement some structure.
- The authors use likelihood and risk in describing the results – it would be good to use odds instead (eg 4.23 times greater odds) and be consistent in the presentation of results.
Discussion
The discussion is well structured according to the study hypotheses.
- Line 86: “3 to 2 times higher”, please change to 2 to 3 times higher odds.
- Line 312: “intoxication among alcohol users” – I’m not sure how this fits in. Few patients would be admitted with intoxication only, I assume - possibly with injuries. Those admitted with intoxication may also be more likely to be hazardous drinkers and not the group accessing addiction treatment centres.
- The paragraph 341-357 was less structured, but it seems that this study also aimed to identify vulnerable groups within an already vulnerable population and maybe this discussion could be framed in this way. A couple of statements in this paragraph seemed to go beyond the scope of the study and not be well referenced, eg: “Regarding the lower risk of frequent ED use in semi-urban areas compared with urban areas, this may reflect the depersonalization of services in urban areas”
Reviewer 2 Report
This manuscript analyzed the different socildemographic and clinical characeristics and outpatient service use which could predict the frequency of emergency department use and hospitalization status with the patients have substance-related disorders. They found high continuity of physician care protected against acute care use, however lots of other clinical or socialdemographic variables like the polysubstance use or homelessness history may increase the usage of acute care. Overall, this study is including a big number of patients and analyzed different variables of the patients background in detail, however, the novelty and scientific significance is relative low, especially the patients database were from 2015-2016; furthermore, some of the results are confused and not support their conclusions, please explain.
- the database for the patients are from 2015-2016, which is more than 6 years old, is there any specific reason to analyzed the study which collected so many years ago? Is it truelly reflect the patients situation currently?
- the last paragraph of the results was confusing. The manuscript described the high continuity of the physican use could protect against the acute care use, however, in the last paragraph of results rection, authors saying that the use of either GP/psychiatrist will increase the frequency of ED use. Please explain.
Round 2
Reviewer 2 Report
The authors answered all my questions and I don't have further questions.